# Health visiting in the UK in light of the COVID-19 pandemic experience (RReHOPE): a realist review protocol

Emma King ,[1] Erica Gadsby ,[1] Madeline Bell,[2] Claire Duddy ,[3] Sally Kendall ,[4] Geoff Wong [3]

¹Faculty of Health Sciences and Sport, University of Stirling, Stirling, UK
²PPI Lead, No institution, Canterbury, UK
³Nuffield Department of Primary Care Health Sciences, University of Oxford, Oxford, UK
⁴Centre for Health Services Studies, University of Kent, Canterbury, UK

**Correspondence to**
Dr Emma King;
emma.king@stir.ac.uk

## ABSTRACT

**Introduction** Health visiting services, providing support to under 5s and their families, are organised and delivered in very different ways in different parts of the UK. While there has been attention to the key components of health visiting practice and what works well and how, there is little research on how health visiting services are organised and delivered and how that affects their ability to meet their objectives. The COVID-19 pandemic rapidly disrupted service delivery from March 2020. This realist review aims to synthesise the evidence on changes during the pandemic to identify the potential for improving health visiting services and their delivery.

**Methods and analysis** This review will follow the RAMESES (Realist And Meta-narrative Evidence Syntheses: Evolving Standards) quality standards and Pawson's five iterative stages to locate existing theories, search for evidence, select literature, extract data, synthesise evidence and draw conclusions. It will be guided by stakeholder engagement with practitioners, commissioners, policymakers, policy advocates and people with lived experience. This approach will consider the emerging strategies and evolving contexts in which the services are delivered, and the varied outcomes for different groups. A realist logic of analysis will be used to make sense of what was happening to health visiting services during and following the pandemic response through the identification and testing of programme theories. Our refined programme theory will then be used to develop recommendations for improving the organisation, delivery and ongoing postpandemic recovery of health visiting services.

**Ethics and dissemination** General University Ethics Panel approval has been obtained from University of Stirling (reference 7662). Dissemination will build on links to policymakers, commissioners, providers, policy advocates and the public. A range of audiences will be targeted using outputs tailored to each. A final stakeholder event focused on knowledge mobilisation will aid development of recommendations.

**PROSPERO registration number** CRD42022343117.

## BACKGROUND

Child health programmes (CHPs) in the UK are universal in reach and offer every family an evidence-based programme of interventions and guidance to support parenting and healthy choices. Broadly, they aim to reduce inequalities and risk, ensure readiness for school, support autonomy and independence, and increase life chances and opportunity.[1] The early years element (0–5) of CHPs is led by health visitors—registered nurses/midwives with additional training in community public health nursing. Health visiting teams include a variety of practitioners, such as community staff nurses and nursery nurses, and link closely with school nurses who lead the 5–19 element of the programme. While the objectives of the universal CHPs across the four nations of the UK are similar, there is considerable variation in how they are organised and delivered.[2] For example, there is variation in the number and intensity of core contacts between families and services, where those contacts may take place and who should undertake them.[3] While models of health visiting can be found elsewhere, the policy contexts influencing service commissioning, delivery, organisation and nursing/professional education differ significantly across countries.[4]

### STRENGTHS AND LIMITATIONS OF THIS STUDY

⇒ Stakeholder and lived experience input will be integral to every part of this study, including developing the focus of the review, identifying evidence, refining programme theory and 'sense-checking' recommendations.
⇒ The review will go beyond published data to include grey literature such as reports, policy documents and other unpublished evidence.
⇒ This realist review will follow current accepted quality and publication standards and methodological practice.
⇒ A realist logic of analysis will be used to develop propositions that can be justified with evidence.
⇒ This review will limit sources to those written in English and will maintain a UK focus, so examples of early years health visiting in other countries may be excluded.

There is little evidence to suggest how best to deliver CHPs to different groups and in different contexts. Health visiting services are complex and highly relational; partnership, integration, communication, and multiagency work are key to service delivery.[5] We know that this service organisation and delivery affect how likely health visiting services are to succeed in their public health goals,[4 6 7] but we do not know much about how, why or in what contexts.[8]

The pandemic has provided an invaluable opportunity to reflect on the different ways that health visiting services are organised and delivered. Health visiting services were affected by the UK Government's Coronavirus Action Plan and similar plans from the devolved governments. Some critical services were not delivered and very few parents of under 2s were able to see a health visitor face-to-face.[9–11] The picture varies notably between UK countries and localities, with multiple examples of differences in the local application of COVID-19 safety rules and resource prioritisation.[10 12 13]

What happens in health visiting has sometimes been referred to as a 'black box' of complexities.[14–16] While there is undoubted value in health visiting, 'the what and the how' (what happens inside that black box) is less clear. The COVID-19 pandemic might be conceptualised as a natural experiment which forced rapid and dramatic changes to the way health visiting was delivered. From this, new data are emerging that might help to reveal explanations of why, how and when certain outcomes occur. This presents an opportunity, therefore, to learn new things about the service in relation to what works for whom in what circumstances.

There has been a proliferation of rapid studies examining COVID-related healthcare services in general,[17] paediatric healthcare,[18] and health visiting services at local and national levels.[11 13 19–21] The pandemic brought about rapid and significant changes to delivery of services, often with little preparation, training, or evidence of effectiveness.[11 13] The pandemic also brought an opportunity to rethink roles, responsibilities and service provision.[20] These changes can be investigated to better understand how practitioners work effectively with families and individuals in different contexts,[22] to identify ways to improve health visiting services and to understand how to improve service recovery after the pandemic. Early child development has significant potential to affect health inequalities across the whole of society.[23 24] The care given to a child, especially during the 1001 critical days (from conception to age 2), has a significant impact on the health, wellbeing and opportunities of that child throughout life.[25]

We will use a realist review to understand how these two issues—the pandemic response and the organisation and delivery of UK health visiting services—impacted on each other, how, why, in what contexts, to what extent and for whom. Realist review goes beyond a standard systematic review to identify the underlying causal mechanisms of complex interventions.[26] Within realist reviews, mechanisms are conceptualised as hidden, context-sensitive, causal forces. Mechanisms may be found at different 'levels' (eg, organisation or individual level), but to make the claim that something is functioning as a mechanism, it is important to consider for which outcome this claim is being made. Importantly, mechanisms are not merely an intervention, an activity, or an intervention component.[27 28] Hence, realist reviews will produce knowledge claims about causes of outcomes in certain contexts using the context–mechanism–outcome configuration (CMOC) heuristic, as this links the three main concepts of a realist analysis.

This realist review approach will help to make sense of what was happening to health visiting services during and following the pandemic response, particularly in understanding circumstances under which it was more or less likely to affect outcomes for children and families, through the identification and testing of programme theories. The rapidly changing context during COVID will allow us to investigate the programmatic assumptions underlying health visiting because so many of these accepted ways of working were changed during the COVID-19 pandemic response. From this, we anticipate recommendations for service improvement to emerge.

### Aim

The aim of this study is to understand the ways in which the COVID-19 pandemic has impacted on health visiting services in the UK in order to identify how the organisation and delivery of health visiting services can be improved for a stronger postpandemic recovery in service delivery. This will be done by means of a realist review of the literature and with key stakeholder engagement across the UK.

The study seeks to answer the question: how can the organisation and delivery of health visiting services in the UK be improved in light of the COVID-19 pandemic, to provide equitable, effective and efficient services for young children and their families?

### Objectives

► To conduct a realist review of the literature to examine what the impacts (both positive and negative) of the COVID-19 pandemic have been on health visiting services in the UK, for whom, in different contexts.

► To engage with key policy, practice and research stakeholders in England, Scotland, Wales and Northern Ireland to understand important contextual differences across the UK in relation to the planning, organisation and delivery of health visiting services.

► To identify recommendations for improving the organisation and delivery and ongoing postpandemic recovery of health visiting services in different settings, for different groups.

The project will run from 1 June 2022 to 30 November 2023.

## METHODS AND ANALYSIS

The protocol here has emerged from previous work by Gadsby *et al*,[18] which examined the impact of the COVID-19 pandemic response on paediatric services in Scotland and England, and by ongoing engagement with the Institute of Health Visiting's work on examining impact. Stakeholder engagement is essential in this project, given the rapidly changing context, the importance of local knowledge and unpublished materials in this review, and the complexity of differences and similarities in health visiting organisation and delivery across the UK. Approximately 25 professionals (policy leads, commissioners, practitioners, and policy advocates) will make up our stakeholder group, with representatives from Northern Ireland, Scotland, Wales and England. This number is manageable from a practical point of view, yet will ensure we can include appropriate expertise by mapping stakeholders onto a grid of five professional areas (policy, commissioning, practice, academia and advocacy) covering five geographical areas (Northern Ireland, Scotland, Wales, England and UK-wide).[29] Stakeholders will be approached through targeted sampling and snowball sampling. This group will meet six times throughout the 18 months of the study. Stakeholder representatives will help us to define the focus of the review, identify evidence for inclusion, refine our programme theory and 'sense check' our recommendations.

### Patient and public involvement

In the design and drafting of this study protocol, engagement with parenting groups/networks and new parents who have encountered health visiting services during the pandemic, and insights from our patient and public involvement lead, helped to inform our realist approach and research questions. For the duration of the study, a separate group of eight people with lived experience (who have had cause to access health visiting services during the pandemic period) will work with us, alongside the professional stakeholder group. Our target of eight lived experience representatives was chosen to allow two representatives from each of the different countries (Northern Ireland, Scotland, Wales and England) while still being a small enough group to allow for meaningful discussion and contribution during online meetings. Potential representatives will be sought through advertising on social media and press releases. Where possible, we will purposively select members of the group to achieve a mixed group profile, for example, based on where they live, how many children they look after and ethnicity. This group will meet four times throughout the 18 months of the study.

The involvement of both professionals and people with lived experience in the design, conduct and dissemination of this work will help to ensure this study delivers new knowledge, meaningful recommendations and accessible outputs that meet the needs of patients and the wider public, commissioners, providers and policymakers. No patients will be directly involved as research participants in this study.

### Realist review stages

The plan of investigation will follow this protocol, which is informed by Pawson's five iterative stages in realist reviews.[30] This process of explanation building starts with the development and refinement of a realist 'programme theory' of UK Health Visiting during the pandemic. This initial programme theory will be refined (see step 1) and then further refined and tested against empirical evidence during the review (see steps 2–5).

#### Step 1: locating existing theories

This step entails locating underlying programme theories for health visiting service delivery during the COVID-19 pandemic, to understand how health visiting services might have been affected, in what ways and with what consequences. Early discussions and literature scoping have informed an initial programme theory (see figure 1), showing the possible contexts, mechanisms and outcomes of interest. If we assume that the mechanisms by which health visiting outcomes are achieved take effect in certain activating contexts, then the changes in service organisation and delivery prompted by the pandemic might well reveal new understandings of these contexts and mechanisms. The changes in context may reveal explanations of why, how and when certain outcomes occur.

This will be further developed in step 1, in collaboration with our stakeholder groups, so that it can be tested and refined in later steps. Content experts in our professional stakeholder group and informal searches of published literature and current policy documents will help to identify existing theories. This informal searching is exploratory and aimed at quickly identifying the range of possible explanatory theories that may be relevant, using exploratory search methods, such as citation tracking and snowballing,[31] along with more structured searching for theories.[32]

#### Step 2: search for evidence

To further develop and refine the programme theory from step 1, a formal evidence search will be conducted. The search strategy will combine general terms describing health visitors and health visiting services, and terms referring to specific UK-based programmes and policies, with terms describing the COVID-19 pandemic (online supplemental file 1). Searches will be limited to identify literature published from 2020 onwards (to coincide with the start of the COVID-19 pandemic response in the UK).

The search strategy will be designed, piloted and conducted by an experienced librarian (CD) in collaboration with the rest of the project team. In addition to database searches (in CINAHL, MEDLINE, Embase, HMIC and Google Scholar), relevant organisation websites will be searched and stakeholders will be consulted to identify grey literature such as reports, policy documents and

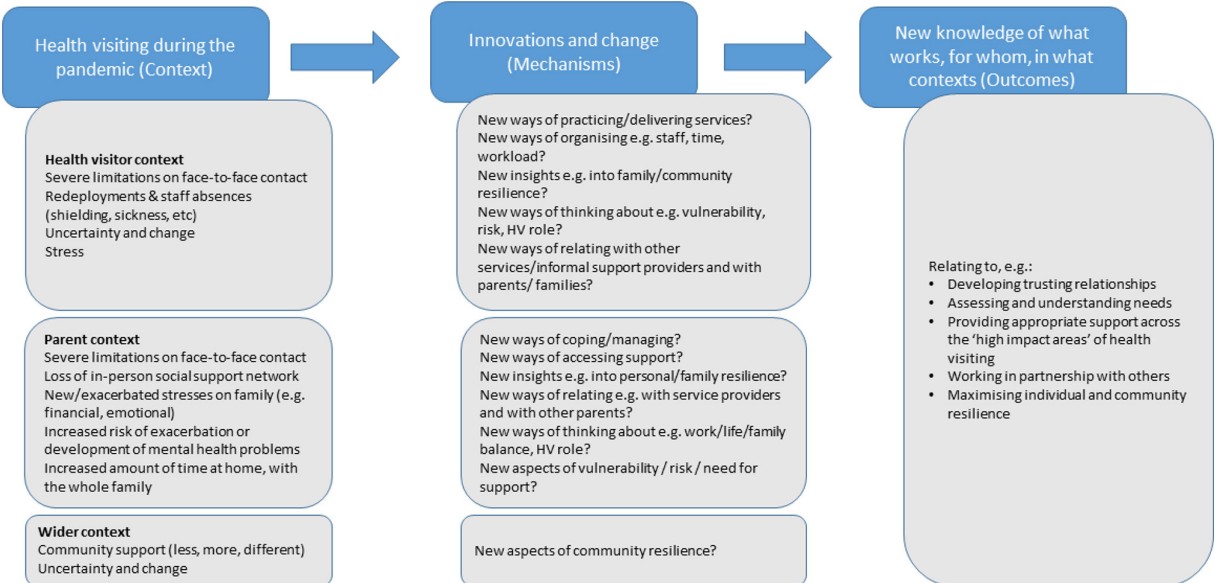

**Figure 1** Initial programme theory with possible contexts, mechanisms and outcomes of interest.

other unpublished evidence. 'Cited by' searches using Google Scholar, and the screening of reference lists of included documents will help to ensure key references are not missed. Given the iterative nature of realist reviews, search terms will be refined, and additional searches will be conducted as required, to develop and test certain subsections of the programme theory.

### Screening
Screening will be undertaken first against title and abstract and then by full text by the research fellow (EK). At both stages, a 10% random subsample of the citations retrieved from searching will be reviewed independently by the co-principal investigators (PIs) (eg, EG and SK) for quality control. Disagreements not resolved through discussion between the three researchers will be resolved through majority vote within the research team. Criteria for initial screening will be broad to ensure all potentially relevant evidence is included:

### Inclusion
► Type of intervention: health visiting programme.
► Study design: all study designs.
► Types of settings: any setting providing health visiting services.
► Types of participants: all families eligible for health visiting services.
► Outcome measures: all outcome measures related to health visiting services.

### Exclusion
Health visiting type models or programmes run in countries other than England, Wales, Scotland and Northern Ireland.

For further exploratory and purposive searches, more specific inclusion and exclusion criteria will be established by the team. This is what Booth *et al* has termed the 'mushroom' method of searching, combining an overall search (the mushroom cap) with additional specific search(es) (the mushroom stalk).[33] This review looks specifically at the UK health system, to best inform the delivery of UK CHPs within their specific political and historical contexts. We may expand searches to include literature on services closely aligned with health visiting, for example, police or social work. We may also expand our criteria to look at similar roles in other countries, although health visiting in its practice orientation to prevention and health promotion for the 0–5 year age group is rarely observed in other countries despite the provision of other child health nursing models.[4]

### Step 3: article selection
Full-text documents will be selected for inclusion in the review based on an assessment of their relevance and rigour.[34] Documents will be selected for inclusion when they contain data that could inform some aspect of the programme theory. At the point of inclusion based on relevance, an assessment will also be made of rigour (how trustworthy and plausible were methods used to generate the data). Judgements about rigour will be made at the level of the included data (where necessary) and programme theory.[35] A random sample of 10% of documents will be selected and independently assessed by the co-PIs. Decisions to include/exclude will be discussed between the three researchers to ensure they have been made consistently. Any disagreements that are not resolved between the three researchers will be resolved by the team through majority vote. Any uncertainties about relevance and/or rigour in the remaining 90% will be treated in the same way: first through discussion with the co-PIs then, if necessary, resolved by the team.

### Step 4: extracting and organising data
Data extraction and data organisation will be undertaken by the research fellow (EK), with a random sample of 10%

independently checked by the co-PIs (eg, EG and SK) for quality control, with disagreements handled in the same way as above. The characteristics of the documents will be extracted for descriptive purposes into an Excel spreadsheet. For data analysis, the full texts of the included papers will be reread, then uploaded into NVivo (a qualitative data analysis software tool) and initially thematically coded. This coding will be deductive (to evaluate potential propositions included in the IPT), inductive (to enable new ideas and propositions to emerge from the evidence) and retroductive (to identify and explore patterns, using theory to offer causal explanations). Refinements of codes (or theories) will be documented in attached memos. Interpretations and judgements will be subsequently confirmed with the rest of the team, and at key points, with our stakeholder groups. As refinements are made, included studies will be rescrutinised to search for data relevant to the revised theory that may have been missed initially.

### Step 5: synthesising the evidence and drawing conclusions

A realist logic of analysis will be used to analyse the extracted data, moving between evidence and theory to develop and refine explanations about why certain patterns are occurring. Throughout the analytic process, propositions, justified with evidence, will be configured to help conceptualise underlying generative mechanisms. These propositions will be represented through CMOCs to describe how specific contextual factors (C) work to trigger particular mechanisms (M), to generate various outcomes (O).[36] Our stakeholder groups will help provide a richer understanding of contexts and mechanisms in different localities.

Our process of analysis and synthesis will be guided by the following questions, which will be asked in relation to each data source.[37]

► Relevance: are sections of text relevant to programme theory development?
► Rigour: are these data sufficiently trustworthy to warrant making changes to any aspect of the programme theory?
► Interpretation of meaning: if the section of text is relevant and trustworthy enough, do its contents provide data that may be interpreted as functioning as context, mechanism or outcome?
► Interpretations and judgements about CMOCs: what is the CMOC (partial or complete) for the data that has been interpreted as functioning as context, mechanism or outcome? Are there further data to inform the particular CMOCs contained within this document or other documents? If so, which other documents? How does this particular CMOC relate to other CMOCs that have already been developed?
► Interpretations and judgements about programme theory: how does this particular (full or partial) CMOC relate to the programme theory? Within this same document, are there data which inform how the CMOC relates to the programme theory? If not, are

there data in other documents? Which ones? In light of this particular CMOC and any supporting data, does the programme theory need to be changed?

Not all parts of a CMOC configuration will be present in the same document, so synthesising data from different documents will be necessary to inform our interpretation of the relationships between contexts, mechanisms and outcomes.

When working through the questions set out, where appropriate, we will use the following forms of reasoning to make sense of the data:

► Juxtaposition of data: for example, where data about how the pandemic influenced outcomes in the universal health visiting programme in one document enable insights into data about outcomes in another document.
► Reconciling of data: where data differ in apparently similar circumstances, further investigation is appropriate in order to find explanations for why these differences have occurred.
► Adjudication of data: on the basis of methodological strengths or weaknesses.
► Consolidation of data: where outcomes differ in particular contexts, an explanation can be constructed of how and why these outcomes occur differently.

## ETHICS AND DISSEMINATION

We will discuss emerging findings and sense check and further refine recommendations with our stakeholder groups. Our dissemination strategy will build on an integrated approach with input from key stakeholders and particularly the Institute of Health Visiting. We will target outputs at a range of audiences, including policymakers and commissioners of health visiting services, providers of health visiting and early years services, and members of the public.

**Acknowledgements** We thank the members of our stakeholder and lived experience groups for their ongoing involvement in this study.

**Contributors** EK: methodology, writing (original Draft, review and editing); EG: conceptualisation, methodology, funding acquisition and supervision in writing (review and editing); MB: conceptualisation, lived experience engagement and writing (review and editing); CD: conceptualisation, methodology, funding acquisition, writing (original draft, review and editing) and data curation; SK and GW: conceptualisation, methodology, funding acquisition, writing (review and editing).

**Funding** This study is funded by the National Institute for Health and Care Research(NIHR) Health and Social Care Delivery Research Programme (NIHR134986).

**Disclaimer** The views expressed are those of the authors and not necessarily those of the National Institute for Health and Care Research or the Department of Health and Social Care.

**Competing interests** None declared.

**Patient and public involvement** Patients and/or the public were involved in the design, conduct, reporting or dissemination plans of this research. Refer to the Methods and analysis section for further details.

**Patient consent for publication** Not applicable.

**Provenance and peer review** Not commissioned; externally peer reviewed.

**ORCID iDs**
Emma King http://orcid.org/0000-0003-3611-9647
Erica Gadsby http://orcid.org/0000-0002-4151-5911
Claire Duddy http://orcid.org/0000-0002-7083-6589
Sally Kendall http://orcid.org/0000-0002-2507-0350
Geoff Wong http://orcid.org/0000-0002-5384-4157

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
