## [Reviewer comments · BMJ Open]

ARTICLE DETAILS

TITLE (PROVISIONAL)	Health visiting in light of the COVID-19 Pandemic Experience (RReHOPE): A realist review protocol
AUTHORS	King, Emma; Gadsby, Erica; Bell, Madeline; Duddy, Claire; Kendall, Sally; Wong, Geoff

VERSION 1 – REVIEW

REVIEWER	Contandriopoulos, Damien University of Montreal, Faculty of Nursing
REVIEW RETURNED	03-Jan-2023

GENERAL COMMENTS	The manuscript is a study protocol for a realist review of the learnings that can be gathered from the disruptions caused by the pandemic to health visiting in the UK. My assessment of the study protocol is unenthusiastic. I feel there are some serious logical and methodological flaws in what is put forward here. The reviewing instructions state: “For studies that are ongoing, it is generally the case that very few changes can be made to the methodology. As such, requests for revisions are generally clarifications for the rationale or details relating to the methods. If there is a major flaw in the study that would prevent a sound interpretation of the data, we would expect the study protocol to be rejected.” In this context, I don’t support the publication. More detailed comments are presented below. Page 5 line 22: The authors state that “the pandemic has provided an invaluable opportunity” for research but the rationale of why this is the case is unconvincing. Yes, the pandemic brought some disruption and variability in the way services were delivered. But how those elements will be harnessed as a learning opportunity is mostly left for the reader to imagine. Page 6 line 42: The research questions would be clearer if the nature of what is meant by “mechanisms” were specified. To me the first question, “What are the mechanisms that explain variation in and mitigation of impacts of the COVID- 19 pandemic in different contexts?” mostly makes sense if “mechanism” is understood at the organizational level. What explains why some delivery organizations were able to adjust and adapt wisely while some other did not? However, in the second question, “What are the important contexts that influence whether the different mechanisms produce the outcomes that have been identified in the literature?” the term “mechanism” mostly makes sense if understood at the production level. How does the interaction of structures and processes influence outcomes? I have a hard time
--

	figuring out an interpretation of the term “mechanism” that could make sense in both the first and second questions. The centrality of those “mechanisms” in the overall project really calls for more clarity in the phrasing here. Page 7 line 16: More details on how will the stakeholders be identified and what sampling or inclusion logic is underlying those choices would be helpful. The same is true for patients (line 37 and up). Why 25 stakeholders? Why 8 patients? How and why did the team chose those numbers? Page 8 line 23: The authors refer to Figure 1 as summarizing a first iteration of a “program theory” of HVS during COVID. However, my understanding of that figure is that it provides little to nothing in terms of listing and organizing factors that had an impact on service delivery. At this point, I’m not sure it makes sense to suggest that this figure provides a program theory that can be tested and refined. Page 9 line 4: I agree that realist review literature searches are often iterative. However even the most iterative search still has to start somewhere. Providing no clue whatsoever of the keywords / approaches that will be used to launch the evidence search is quite unconvincing. Same for screening (line 11), stating that screening will be run first on abstracts and then on full texts but without giving a hint of a clue about the screening criteria left me with the feeling this section of the proposal is severely undeveloped. Page 9 line 26: The inclusion and exclusion criteria are basically convoluted ways to mention that the search will be focused on the UK. Beyond the fact that the current phrasing is, in my view, suboptimal, I think a stronger rationale of why the study will only look at literature from the UK would be needed. My experience with realist reviews is that causal pathways informing the program theory can be (and often are) derived from other jurisdictions. If the authors believe this won’t be the case here, they should provide a justification of why that would be the case. Page 10 line 50: The authors previously justified the fact they were not providing any clues on what search approach, keyword or syntax would be used by the fact the evidence search was expected to be highly iterative. However I have a hard time figuring out how the selection and synthesis approach put forward here will accommodate this “iterativity.” My understanding of an iterative approach to knowledge synthesis is that the analysis and data collection are conducted either simultaneously or in multiple sequential steps. In any case, the ongoing analysis informs the search in the same way that the search informs the analysis. However I can’t see how the methods / coding put forward here will accommodate such a dialogic approach. Page 13 line 27: The authors mentioned that they plan to collaborate with a large group of patients and stakeholders. This is reiterated here but more details on the how / when and why would be needed in order for the reader to understand the nature and extent of that collaborative model. As a minor comment, the reviewing instructions state that “protocol papers should report planned or ongoing studies. The dates of the
--	--

	study should be included in the manuscript.” And no date nor timeframes are provided here.
--	--

VERSION 1 – AUTHOR RESPONSE

Reviewer: 1

Prof. Damien Contandriopoulos, University of Montreal Comments to the Author:

The manuscript is a study protocol for a realist review of the learnings that can be gathered from the disruptions caused by the pandemic to health visiting in the UK.

My assessment of the study protocol is unenthusiastic. I feel there are some serious logical and methodological flaws in what is put forward here. The reviewing instructions state: “For studies that are ongoing, it is generally the case that very few changes can be made to the methodology. As such, requests for revisions are generally clarifications for the rationale or details relating to the methods. If there is a major flaw in the study that would prevent a sound interpretation of the data, we would expect the study protocol to be rejected.” In this context, I don’t support the publication. More detailed comments are presented below.

Page 5 line 22: The authors state that “the pandemic has provided an invaluable opportunity” for research but the rationale of why this is the case is unconvincing. Yes, the pandemic brought some disruption and variability in the way services were delivered. But how those elements will be harnessed as a learning opportunity is mostly left for the reader to imagine.

Thank you for pointing out that the rationale for this study was not clear enough in our first submission. In our Background section, we have made clearer the opportunity presented by the COVID pandemic to look inside the ‘black box’ of health visiting, to better understand the mechanisms by which outcomes are achieved in different contexts. Understanding the assumptions and how this affected the mechanisms is what will contribute to the learning opportunities.

Page 6 line 42: The research questions would be clearer if the nature of what is meant by “mechanisms” were specified. To me the first question, “What are the mechanisms that explain variation in and mitigation of impacts of the COVID- 19 pandemic in different contexts?” mostly makes sense if “mechanism” is understood at the organizational level. What explains why some delivery organizations were able to adjust and adapt wisely while some other did not? However, in the second question, “What are the important contexts that influence whether the different mechanisms produce the outcomes that have been identified in the literature?” the term “mechanism” mostly makes sense if understood at the production level. How does the interaction of structures and processes influence outcomes? I have a hard time figuring out an interpretation of the term “mechanism” that could make sense in both the first and second questions. The centrality of those “mechanisms” in the overall project really calls for more clarity in the phrasing here.

Our review questions have been through internal peer reviews and have also been approved by our funder NIHR and their two external peer reviewers. We are therefore unable to change the wording of our review questions at this stage. To provide more clarity for readers less familiar with realist approaches, we have made the decision to remove these review questions and to replace them with our research question and objectives.

We have also taken on board your comments that our use of the word ‘mechanisms’ was not clear and added a more detailed explanation of this in our Background section. Bearing in mind that the word ‘mechanism’ is part of the language and a key element of any realist synthesis.

Page 7 line 16: More details on how will the stakeholders be identified and what sampling or inclusion logic is underlying those choices would be helpful. The same is true for patients (line 37 and up). Why 25 stakeholders? Why 8 patients? How and why did the team chose those numbers?

We understand that our current reasons for selecting these numbers were unclear and have clarified our choosing of 25 stakeholders, to represent five professional areas covering five geographical areas. Similarly, 8 lived experience participants were chosen to represent two from each UK country, without the group becoming too big for meaningful engagement. We have made this clearer in the text and briefly explained our sampling strategy in our Methods and Analysis section.

Page 8 line 23: The authors refer to Figure 1 as summarizing a first iteration of a “program theory” of HVS during COVID. However, my understanding of that figure is that it provides little to nothing in terms of listing and organizing factors that had an impact on service delivery. At this point, I’m not sure it makes sense to suggest that this figure provides a program theory that can be tested and refined.

Our study is not seeking to list or organise factors that have an impact on service delivery, and we have clarified this further in the text.

Our initial program theory diagram is a way of representing the context of health visiting that can be shared with our stakeholder groups for further discussion and refinements. Many realist review protocols do not include the initial programme theory, but we believe it is important as the starting step of our study. We feel it usefully represents the possible contexts, mechanisms and outcomes of interest and is comparable with other initial program theory diagrams for realist review protocols published in BMJ Open (Power et al. 2019; Alghamdi et al. 2022; Siersbaek et al. 2022).

Page 9 line 4: I agree that realist review literature searches are often iterative. However even the most iterative search still has to start somewhere. Providing no clue whatsoever of the keywords / approaches that will be used to launch the evidence search is quite unconvincing. Same for screening (line 11), stating that screening will be run first on abstracts and then on full texts but without giving a hint of a clue about the screening criteria left me with the feeling this section of the proposal is severely undeveloped.

We understand that differing studies employ a different approach to this, with some including no keywords (Brennan et al. 2014; Weetman et al. 2017; Power et al. 2019; Alghamdi et al. 2022), some including their initial search strategies in a supplementary file (Shahid al. 2021; Evans et al. 2022) , and others occasionally including a search strategy in the main text. (Siersbaek et al. 2022; Finlay-Jones 2021).

Taking on board our reviewer’s comments we have given more detail of our proposed search strategy in our ‘locate existing theories’ section. We have also proposed a supplementary file with more detail of the key-words of our proposed searches at this protocol stage. As the focus and scope of our searches will be informed by our stakeholder group, we feel it would be inappropriate to publish our full search strategy before these voices are heard. We plan to publish our detailed literature searches at a later point in the study.

Page 9 line 26: The inclusion and exclusion criteria are basically convoluted ways to mention that the search will be focused on the UK. Beyond the fact that the current phrasing is, in my view, suboptimal, I think a stronger rationale of why the study will only look at literature from the UK would be needed. My experience with realist reviews is that causal pathways informing the program theory can be (and often are) derived from other jurisdictions. If the authors believe this won’t be the case here, they

should provide a justification of why that would be the case.

In our Introduction we have made clearer the reasons for focusing on the UK and that the issues are very specific to the UK health system. This is due to the organisation of regional areas across four countries within the UK, and the nature of this being publicly funded. COVID has had a particularly unique impact on a system that aims to provide a public health and preventative service designed for children aged 0-5.

We have also clarified that we may expand our search to look at other organisations within the UK who are closely aligned with Health Visiting (e.g. police, social work) if necessary.

Page 10 line 50: The authors previously justified the fact they were not providing any clues on what search approach, keyword or syntax would be used by the fact the evidence search was expected to be highly iterative. However I have a hard time figuring out how the selection and synthesis approach put forward here will accommodate this “iterativity.” My understanding of an iterative approach to knowledge synthesis is that the analysis and data collection are conducted either simultaneously or in multiple sequential steps. In any case, the ongoing analysis informs the search in the same way that the search informs the analysis. However I can’t see how the methods / coding put forward here will accommodate such a dialogic approach.

Within realist reviews, there are no pre-specified or obligatory ways for how to operationalise the analysis of data and also whether additional searches are needed.

Most reviews run an initial larger formal search and then proceed to analyse these documents. Should the analyses indicate ‘gaps’ in the data needed to confirm, refute or refine programme theory, then additional searches may be run. This is what Booth (Booth et al. 2020 Research Synthesis Methods) has termed the ‘mushroom’ approach, and we have added an explanation of this to our Screening section. (Carrieri D et al. Health Serv Deliv Res 2020;8(19);Price T et al Health Serv Deliv Res 2021;9(11); Papoutsis C et al. Health Serv Deliv Res 2018;6(10))

Page 13 line 27: The authors mentioned that they plan to collaborate with a large group of patients and stakeholders. This is reiterated here but more details on the how / when and why would be needed in order for the reader to understand the nature and extent of that collaborative model.

We assume this might refer to page 12 in the proof copy supplied to us, which was formerly in the discussion which has now been removed. We have however sought to clarify this at an earlier stage.

As a minor comment, the reviewing instructions state that “protocol papers should report planned or ongoing studies. The dates of the study should be included in the manuscript.” And no date nor timeframes are provided here.

Thank you for spotting this omission, which has now been rectified and dates added to the ‘Objectives’ section.

Reviewer: 1

Competing interests of Reviewer: No competing interests to report